# Reactions with Proteins of Three Novel Anticancer Platinum(II) Complexes Bearing N-Heterocyclic Ligands

**DOI:** 10.3390/ijms221910551

**Published:** 2021-09-29

**Authors:** Francesca Sacco, Matteo Tarchi, Giarita Ferraro, Antonello Merlino, Giorgio Facchetti, Isabella Rimoldi, Luigi Messori, Lara Massai

**Affiliations:** 1Department of Chemistry, University of Florence, Via della Lastruccia 3-13, 50019 Sesto Fiorentino, Italy; francesca.sacco@unifi.it (F.S.); matteo.tarchi@unifi.it (M.T.); lara.massai@unifi.it (L.M.); 2Department of Chemical Sciences, University of Naples Federico II, Via Cintia, 80126 Napoli, Italy; giarita.ferraro@unina.it (G.F.); antonello.merlino@unina.it (A.M.); 3Department of Pharmaceutical Sciences, Università degli Studi di Milano, Via Venezian 21, 20133 Milano, Italy; giorgio.facchetti@unimi.it

**Keywords:** platinum complexes, mass spectrometry, crystallography

## Abstract

Three novel platinum(II) complexes bearing N-heterocyclic ligands, i.e., Pt2c, Pt-IV and Pt-VIII, were previously prepared and characterized. They manifested promising in vitro anticancer properties associated with non-conventional modes of action. To gain further mechanistic insight, we have explored here the reactions of these Pt compounds with a few model proteins, i.e., hen egg white lysozyme (HEWL), bovine pancreatic ribonuclease (RNase A), horse heart cytochrome c (Cyt-c) and human serum albumin (HSA), primarily through ESI MS analysis. Characteristic and variegate patterns of reactivity were highlighted in the various cases that appear to depend both on the nature of the Pt complex and of the interacting protein. The protein-bound Pt fragments were identified. In the case of the complex Pt2c, the adducts formed upon reaction with HEWL and RNase A were further characterized by solving the respective crystal structures: this allowed us to determine the exact location of the various Pt binding sites. The implications of the obtained results are discussed in relation to the possible mechanisms of action of these innovative anticancer Pt complexes.

## 1. Introduction

Tumor pathologies remain one of the major causes of disease of the new millennium [1]. Great achievements have been obtained through modern and personalized therapies for many types of cancers. Unfortunately, however, there is a category known as “orphan tumors” that certainly represents one of the greatest research challenges [2]. These tumors are characterized not only by being very aggressive but also by the fact that they are “orphans” of therapy. Among them, triple negative breast cancer (TNBC) is certainly one of the most aggressive [3]. For the treatment of this type of tumor, therapy based on cisplatin and its derivatives still represents the only therapeutic protocol currently in use, despite the severe side effects and limits of Pt drugs such as neurotoxicity and a high frequency of Pt resistance [4]. 

In a way, the FDA approval of platinum-based drugs started a new era in the field of metallodrugs, but the persistence of some severe side effects, together with the arising of resistance phenomena, urgently requires a new generation of transition metal-based chemotherapeutics able to overcome these limitations. The identification of intracellular targets for metallodrugs is important for clarifying their possible mechanisms of action, but it is still a major challenge. While DNA has been identified as a privileged target for metal-based organometallic compounds since the very early research works in this field [5], new omics techniques have revealed their great potential in identifying different molecular targets and interaction sites [6]. 

In recent years, in our laboratory, we have developed two different series of cationic platinum complexes having different heterocyclic amines as coordinated ligands (Figure 1) [7,8]. Among the first of these, in which a methylamino-imidazole was coordinated to a platinum(II) metal center bearing an additional imidazole ligand substituted with saturated chains of different lengths, the compound Pt2c was found to exert a cytotoxic activity comparable to cisplatin on the TNBC cell line, i.e., MDA-MB-231, with an IC_50_ of 61.9 μM. This remarkable cytotoxicity could be ascribed to its ability to be inactivated to a lesser extent by the endogenous resistance systems such as GSH and metallothioneins (MTs), both responsible for partial cisplatin inactivation [7]. Moreover, compound Pt2c proved to be less toxic for healthy cells and to have an intrinsic chemical stability (up to 9 days) maintaining the imidazole with the chain on the imidazole nitrogen as an additional ligand. These features made it an ideal candidate for targeted therapy through its loading into mesenchymal stromal cells (MSCs) [9].

We have, thus, synthesized a new series starting from 8-aminoquinoline and its enantiopure derivatives containing instead the tetrahydroquinoline ring. Among the compounds of this series, Pt-IV was found to be the most active on the MDA-MB-231 cancer cell line with an IC_50_ of 10.9 μM. ICP-MS analysis data suggested that the latter complex is taken up more efficiently than cisplatin at the same concentration. Compared to the charged complex Pt2c, Pt-IV interacts with DNA, but to a much lesser extent than cisplatin, and is certainly characterized by a strong interaction with proteins, which suggests a different mechanism of action. 

Conversely, the Pt-VIII complex, which is characterized by the presence of the chiral tetrahydroquinoline ring, was only modestly active, even if an improvement in terms of cytotoxicity with respect to the corresponding dichloride analogue was achieved.

Upon studying the differences between complexes Pt-IV and Pt-VIII, it was observed that both complexes reduce p53 expression and the proliferating cell nuclear antigen (PCNA) and influence α-tubulin and β-actin levels, although Pt-IV is more powerful to induce p53 mRNA than cisplatin. While both Pt-IV and Pt-VIII have the N-hexyl-imidazole additional ligand, they act on two different cell phases, thus underlining the strong impact on the biological profile due to the chemical structure of the chelating diamine. Pt-VIII affects the G2/M phase-like complex Pt2c, while Pt-IV, similarly to cisplatin, interferes with the progression of the G0/G1. The higher cell permeability of these compounds when compared to cisplatin was due to their chemical features, i.e., their overall cationic charge along with the presence of long alkyl chains. In fact, Pt-IV complex reaches a 13-fold higher intracellular concentration than cisplatin, even if it is fully documented that the monofunctional complexes interact modestly with DNA, suggesting a different molecular target [8].

The study of the chemical stability of Pt-IV underlined a possible different mechanism compared to complex Pt2c, as suggested by the greater amount of Pt-IV deactivated through Mets7 interaction after 5 days. Despite these results, Pt-IV as well as Pt2c, can still be considered as new possible candidates for targeted delivery and also in consideration of the ongoing study of the MSCs loading protocol [9,10,11].

Considering the attested different action mechanism of oxaliplatin and its analogues compared to cisplatin and the fact that many other targets are involved in its cytotoxic activity, [12] it emerges that the study of the interactions of platinum compounds with proteins is of great interest for understanding both their mode of action and their toxic effects. For these reasons, we decided to comparatively explore the reactions of the above-mentioned platinum-based compounds with a few model proteins.

Pt compounds/proteins reactions were primarily monitored via ESI MS analysis according to an experimental protocol that has been developed in the laboratory of the University of Florence [10,11]. Additionally, we succeeded in obtaining crystals suitable for X-ray diffraction analysis for the adducts formed by complex Pt2c with RNase A and HEWL. The respective crystal structures were solved.

## 2. Results

### 2.1. ESI MS Results

#### 2.1.1. HEWL

Hen white egg lysozyme (HEWL) has been extensively employed in the MetMed lab in Florence [13,14] for the preliminary interaction studies via ESI MS of metallodrug–protein interactions as it is a small protein with a great ionizability in the ESI source, giving rise to well resolved ESI MS spectra. In addition, HEWL is well prone to crystallization allowing the obtainment of independent crystallographic data on the same systems.

Among the three Pt compounds here analyzed, only complex Pt2c turned out to react with HEWL, producing some interesting adducts. In Figure 2 the deconvoluted mass spectrum of HEWL incubated with complex Pt2c for 48 h at 37 °C is reported: at 14,303 Da the signal of the unbound protein is detected with a relative intensity of 80%, while a series of signals with greater mass values are observed in the 14,600–14,900 Da range. Of these, the first and the most intense signal at 14,759 Da is still not assigned: it is shifted by +456 Da from the unbound protein signal and by −43 Da from the following one. The very intense signal at 14,802 Da is shifted by 499 Da and is attributed to the protein bearing a Pt(II) center and two (1-hexyl-1H-imidazole) moieties. The following signal at 14,832 Da (+529 Da) is ascribed to the protein-bound entire complex Pt2c after the loss of the chloride ligand, i.e., through the fragment [Pt((1-hexyl-1H-imidazol-2yl)(methanamine)(1-hexyl-1H-imidazole)]^2+^. Notably, this latter adduct has the most intense signal observable already after 3 h of incubation, but, over time, the species at 14,759 Da becomes the predominant one, proving to be the most favored adduct.

#### 2.1.2. HSA

Human Serum Albumin (HSA) is the most abundant plasma protein; among the multiple functions it performs, it is responsible for the binding and the transport of different molecules in the body, from fatty acids, vitamins and hormones to exogenous compounds like drugs [15]. The platinum compounds here investigated have been tested with this relevant and significant protein and they show a large tendency to react with it, forming stable adducts over time. Indeed, the three studied Pt compounds invariantly bind the protein after the loss of the labile chloride ligand. Notably, the metalation of the albumin already occurred after 3 h of incubation at 37 °C, for all of the Pt compounds, and the detected adducts were all stable over the following 48 h. In Figure 3 the mass spectra of HSA before and after the incubation with each platinum compound are reported. The deconvoluted mass spectrum of metal-free HSA is characterized mainly by the signals at 66,438 Da and 66,557 Da (Figure 3a) attributed, respectively, to the protein and its cysteinylated form, i.e., that of the protein with a Cys residue bound to the Cys34, affording a typical post-translational modification (PTM) of the protein. The deconvoluted mass spectrum of complex Pt2c with HSA (Figure 3b) shows, beyond the signals of the unbound protein, two additional and very clear signals at greater masses, specifically at 66,966 Da and 67,084 Da. They are attributed to the native protein and its cysteinylated form bound to the Pt-containing fragment [Pt((1-hexyl-1H-imidazol-2yl)(methanamine) (1-hexyl-1H-imidazole)]^2+^. The relative intensities of these signals (over 65%) compared to those of the free protein suggest a high tendency of the protein to react with complex Pt2c.

Pt-VIII reacts with HSA similarly to complex Pt2c: in fact, it gives rise to an adduct with both the native and the cysteinylated protein after the loss of the chloride ligand and through the binding of the Pt-containing fragment, namely [Pt((R)-8-amino-5,6,7,8-tetrahydroquinoline)(1-hexyl-1H-imidazole)]^2+^. The deconvoluted mass spectrum recorded after 48 h (Figure 3c) displays the two signals assigned to the adducts, located at 66,928 Da and 67,050 Da, respectively. Notably, as the relative intensities of these signals are lower than 40%, it can be assumed that the protein is less prone to interact with Pt-VIII than with complex Pt2c, at the same incubation time (48 h).

The Pt-IV compound turned out to be the most reactive. Even at first glance, the deconvoluted mass spectrum recorded after 48 h of incubation of Pt-IV with HSA (Figure 3d) provides evidence for a greater reactivity of this compound with albumin when compared to the other Pt-based drugs here analyzed: indeed, a series of signals at greater masses than that assigned to the free protein is detected. The most intense signals at 66,927 Da and 67,048 Da are ascribed to the native and cysteinylated protein bound to a Pt-containing fragment that corresponds to the Pt-IV molecule without the chloride ligand, i.e., through the fragment [Pt((R)-8-aminoquinoline)(1-hexyl-1H-imidazole)]^2+^. The other two signals, at 67,418 Da and 67,535 Da, are the bis-adducts with the same Pt-containing fragment. Notably, the relative intensity of the signals of the mono-adducts is greater than the protein unbound signals, indicating a high tendency of the protein to interact with this compound.

#### 2.1.3. Cytochrome c

Remarkably, no adduct has been observed in the reaction of Cyt-c with any of the above Pt compounds, upon 48 h of incubation at 37 °C.

#### 2.1.4. RNase A

An interaction study between RNase A and the Pt compounds was also carried out, to help clarify the nature of the complex Pt2c fragment(s) detected from the crystallographic data (see below) and to study the reactivity of the Pt-aminoquinoline compounds with proteins at different incubation times. Beyond the signals of the unbound protein at 13,681 Da and of RNase A with sulphate ion at 13,779 Da, mass spectra reveal that each compound produces one or more signals due to interactions with the protein. These signals are more intense when incubation times are increased, except for the Pt-IV compound, which shows a signal for the adduct at 14,171 Da that does not change its intensity over time. Pt-VIII interacts with RNase A revealing a very clear signal at 14,174 Da, attributed to the adduct of the protein with the whole compound. Interestingly, in the mass spectrum registered in the presence of Pt2c (Figure 4c), three distinct signals emerge; those with a mass of 14,056 Da and 14,208 Da, respectively, are attributed to the adducts of the protein with the fragment [Pt(1-hexyl-1H-imidazol-2-yl) methanamine]^2+^ and with the whole complex Pt2c, while the signal at 14,136 Da, shifted by a +455 Da with respect to the unbound protein, similar to what was already seen with HEWL, remains unassigned.

### 2.2. Crystallographic Results

The X-ray structures of the adducts of RNase A and HEWL with complex Pt2c have been solved at high resolution, 2.13 and 1.22 Å, respectively (Figure 5a,b). The structure of the RNase A adduct was solved using diffraction data collected from crystals with two molecules in the asymmetric unit obtained exposing metal-free protein crystals to a solution of the Pt complex for 12 days (Figure 5a). Similarly, crystals of the adduct of complex Pt2c with HEWL have been obtained exposing crystals of HEWL to a solution of the Pt complex for 12 days.

In both adduct structures, the overall protein conformation is not significantly affected by Pt binding, as frequently observed [16,17]. In fact, root mean square deviations in the positions of the carbon alpha atoms between the structures of the adducts and those of the metal-free proteins are within the range 0.39–0.26 Å in the case of RNase A and 0.23 Å for HEWL. The inspection of the electron density maps allowed the identification of Pt binding sites in both the structures of the adducts. It appears that the Pt compound breaks down upon protein binding and that the Pt center can coordinate to selected protein residues completing its coordination sphere with solvent molecules. In the structure of the adduct formed between complex Pt2c and RNase A, two Pt binding sites were found: close to the side chains of His105 and His119 of both protein molecules in the asymmetric unit. The definition of Pt ligands in this structure is ambiguous and Pt ligands have not been modelled in all of the metal binding sites. This is frequently observed in the X-ray structures of the adducts of the proteins with metallodrugs [18]. In particular, close to His105 of molecule A, a Pt(II) atom coordinates the Nε of the imidazole ring and three water molecules (Figure 6a), while in molecule B, one of the Pt ligands is not defined (Figure 6b). The metal coordination sphere is even less defined close to the side chains of His119. At this Pt binding site in molecule A, the definition of the Pt ligands is complicated by the presence of alternative conformations adopted by the side chain of the His (Figure 6c); in molecule B the metal coordination sphere is completed by a water molecule and two undefined ligands (Figure 6d).

Two Pt binding sites were also observed in the adduct formed between complex Pt2c and HEWL. The first Pt binding site is the side chain of His15 (Figure 6e). Close to this site, which has been identified as a Pt binding site in several previous studies [19,20], the Pt coordination sphere is completed by one water molecule and two unidentified ligands. The second Pt binding site is on the protein surface. Here, the metal center is not coordinated to protein residue side chains. The complex 2c fragment is bound to the protein via stacking interactions formed by one of the Pt ligands, modelled as imidazole in our structure, with the side chain of Trp62 (Figure 6f). The other three Pt ligands are not visible in the electron density map.

In conclusion, crystallographic results indicate that complex Pt2c breaks down under the investigated experimental conditions and that Pt fragments are bound to RNase A and HEWL in the final adduct of the Pt complex with these proteins.

### 2.3. RNase A Activity Studies

To verify that a Pt fragment could effectively bind the side chain of His119, which is a catalytically important residue of RNase A, the enzymatic activity of the adduct of the protein with complex Pt2c on yeast RNA was evaluated using samples of RNase A incubated for different times with the Pt complex.

When RNase A is incubated for 24 or 72 h in the presence of complex Pt2c, the protein loses only a part of its ability to degrade RNA, even at high Pt concentrations (Figure 7a,b). On the contrary, after 12 days incubation, the protein loses more than 50% of its enzymatic activity (Figure 7c). These findings suggest that the Pt complex binds the protein in the active site and that the binding inhibits the protein catalytic activity more efficiently when the incubation time increases.

## 3. Discussion

New pharmaceutically suitable platinum compounds are still intensely searched to find new anticancer drug candidates with a different spectrum of anticancer activity, innovative modes of action and the ability to overcome resistance toward clinically established Pt compounds. The three innovative Pt compounds that are the subject of this investigation appear to manifest promising properties as they conserve a relevant in vitro anticancer activity while exhibiting structural features dissimilar from cisplatin and its analogues and they show a mode of action that is substantially DNA independent.

### 3.1. Protein Interaction Studies through ESI MS Experiments

It is well conceivable that these Pt compounds may induce their cytotoxic effects by interacting primarily with protein targets. To this end, we decided to characterize the general mode of interaction of these Pt compounds with proteins by analyzing in detail their reactions with a few model proteins. ESI MS was used as the election technique for this study while, in selected cases, additional information was gained through the implementation of crystallographic studies.

Four model proteins were used for the ESI MS studies, namely Cyt-c, HEWL, RNase A, and HSA.

ESI MS analysis indicates that the interaction of the three studied compounds with the four model proteins produces a rather complex pattern of metallodrug–protein reactivities. Indeed, no reactivity was observed in the case of Cyt-c; HEWL only reacted with complex Pt2c, while HSA and RNase A tended to react eagerly with all three Pt compounds.

Moreover, the type of resulting adducts was different in the various cases. RNase A favors the formation of a Pt-containing fragment where only the chloride ligand is released. Interestingly, the complex Pt2c tends to form a fragment with a mass of 456 Da, whose nature is still unclear. This fragment probably originates from a rearrangement of an initially formed Pt fragment.

### 3.2. Crystallographic Studies

Crystallographic results were obtained only for the adducts formed between complex Pt2c and the two model proteins HEWL and RNase A. These results permitted localization, with precision, of the major Pt binding sites with these two proteins. Yet they were not able to offer conclusive information on the nature of the other ligands coordinated to the protein-bound Pt center. It is possible that the fragments that are seen in the crystallographic studies can be different from those suggested by the ESI MS studies due to the different experimental conditions and the far longer incubation times of the crystallographic studies.

## 4. Materials and Methods

### 4.1. Materials

Lyophilized HEWL, Cyt-c, RNase A, and HSA were purchased from Sigma-Aldrich and used without further purification or manipulation. Pt compounds, i.e., complex Pt2c, Pt-VIII, and Pt-IV, were synthesized as previously described [7,8]. DMSO was purchased from Fluka. LC-MS grade water, methanol, and ammonium acetate salt were purchased from Sigma-Aldrich.

### 4.2. Electrospray Ionization Mass Spectrometry Analysis: Sample Preparation

Stock solutions of HEWL, Cyt-c, RNase A, and HSA 10^−3^ M were prepared dissolving the lyophilized proteins in LC-MS grade water. Likewise, platinum compound solutions 10^−2^ M were freshly prepared before each experiment solubilizing the powders in LC-MS grade methanol. Then, aliquots of protein stock solution were mixed with aliquots of the platinum compounds at a protein-to-metal compound molar ratio of 1:3, with 2 × 10^−3^ M ammonium acetate solution (pH 6.8) to 10^−4^ M final protein concentration. Each solution was incubated at 37 °C up to approximately 72 h.

### 4.3. Electrospray Ionization Mass Spectrometry Analysis: Final Dilutions

After the incubation time, all solutions were sampled and diluted to a final protein concentration of 10^−7^ M of HEWL and RNase A using 2 × 10^−3^ M ammonium acetate solution (pH 6.8), of 10^−7^ M of Cyt-c using LC-MS grade water, and of 5 × 10^−7^ M of HSA using 2 × 10^−3^ M ammonium acetate solution (pH 6.8). The final solutions were also added with 0.1% *v*/*v* of formic acid just before the infusion in the mass spectrometer.

### 4.4. Electrospray Ionization Mass Spectrometry Analysis: Instrumental Parameters

The ESI mass study was performed using a TripleTOF 5600+ high-resolution mass spectrometer (AB Sciex, Framingham, MA, United States), equipped with a DuoSpray^®^ interface operating with an ESI probe. Respective ESI mass spectra were acquired through direct infusion at 5 μL min^−1^ of flow rate. The general ESI source parameters optimized for each protein were as follows: HEWL positive polarity, ion spray voltage floating 5500 V, temperature 0, ion source Gas 1 (GS1) 40 L min^−1^; ion source Gas 2 (GS2) 0; curtain gas (CUR) 20 L/min, collision energy (CE) 10 V; declustering potential (DP) 200 V, acquisition range 1000–2800 *m*/*z*; Cyt-c positive polarity, ion spray voltage floating 5500 V, temperature 0, ion source Gas 1 (GS1) 40 L min^−1^; ion source Gas 2 (GS2) 0; curtain gas (CUR) 25 L/min, collision energy (CE) 10 V; declustering potential (DP) 100 V, acquisition range 600–2500 *m*/*z*; RNase A positive polarity, ion spray voltage floating 5500 V, temperature 0, ion source gas 1 (GS1) 40 L min−1; ion source gas 2 (GS2) 0; curtain gas (CUR) 15 L min^−1^, declustering potential (DP) 100 V, collision energy (CE) 10 V, range 1000–3000 *m*/*z*; HSA: positive polarity, ion spray voltage floating 5500 V, temperature 25 °C, ion source Gas 1 (GS1) 45 L min^−1^; ion source Gas 2 (GS2) 0; curtain gas (CUR) 12 L min^−1^, collision energy (CE) 10 V; declustering potential (DP) 150 V, acquisition range 1000–2600 *m*/*z*. For acquisition, Analyst TF software 1.7.1 (Sciex) was used, and deconvoluted spectra were obtained by using the Bio Tool Kit micro-application v.2.2 embedded in PeakViewTM software v.2.2 (Sciex).

### 4.5. Crystallization and X-ray Diffraction Data Collection

RNase A and HEWL crystals were grown through the hanging drop vapor diffusion method at 20 °C. A solution containing 22% PEG 4K and 10 mM sodium citrate buffer at pH 5.1 was used as reservoir for RNase A. HEWL crystals grew in the presence of 20% ethylene glycol, 0.1 M sodium acetate buffer pH 4.5 and 0.6 M sodium nitrate. Crystals of the adducts formed upon reaction of complex Pt2c with the two model proteins were obtained by soaking procedure. The Pt complex dissolved in 100% DMSO was added to a drop containing the crystals up to a final concentration of 5 mM, reaching a protein-to-metal compound molar ratio of 1:3.5 for RNase A and of 1:5 for HEWL. X-ray diffraction data on these crystals were collected on a Pilatus detector at XRD-2 beamline of Elettra Synchrotron, Trieste, Italy, after 12 days of soaking. Data were processed using Autoproc [21]. The structures of both adducts were solved by molecular replacement method, using the coordinates deposited in the PDB under the accession codes193L [22] for HEWL and 1JVT [23] for RNase A. Pt binding sites were unambiguously identified by comparing 2Fo–Fc, Fo–Fc, and anomalous difference electron density maps. The structures of the adducts of complex Pt2c with RNase A and HEWL were refined to the R-factor/R-free values of 12.87/16.97 and 19.75/27.43, respectively. The details of crystallographic and refinement parameters are given in Table 1. The refined models and structure factors were deposited in the Protein Data Bank under the accession codes 7PNI (adduct with RNase A) and 7PNH (adduct with HEWL).

### 4.6. RNase A Enzymatic Activity Assays

Enzymatic activity of RNase A was measured by monitoring cleavage of yeast RNA via UV–vis spectroscopy, using the Kunitz method [24]. The protein was incubated for 24 and 72 h and for 12 days in the presence of different concentrations of complex Pt2c to reach final protein–metal ratios of 1:0.5, 1:1, 1:5, and 1:10. After incubation, its activity on yeast RNA was determined at room temperature in 0.050 M sodium acetate pH 5.0, using 0.5 mg mL^−1^ of RNA and enzyme concentration of 0.1 mg mL^−1^. The enzymatic activity of native RNase A was also measured and used as reference. Data reported are average of three independent measurements.

## 5. Conclusions

Base on the grounds of the reported results, we can state that characteristic and variegate patterns of reactivity between the studied Pt complexes and four model proteins take place that appear to depend critically both on the nature of the Pt complex and of the interacting protein. Notably, joint ESI MS and crystallographic studies concur in defining a rather accurate picture of the occurring interactions at the molecular level. Only upon dissecting these modes of interactions and identifying the effective protein targets will it be possible to gain a more precise insight on the respective mechanisms of action of these novel Pt compounds and establish detailed structure–function relationships. As demonstrated by the crystallographic data highlighting the presence of a platinum binding side close to the catalytically active His119 in RNase, in the future it will be interesting to evaluate the inhibitory activity effects produced by Pt compounds, and by Pt2c in particular, on the protein transcription process and their mechanistic importance.

## Figures and Tables

**Figure 1 ijms-22-10551-f001:**
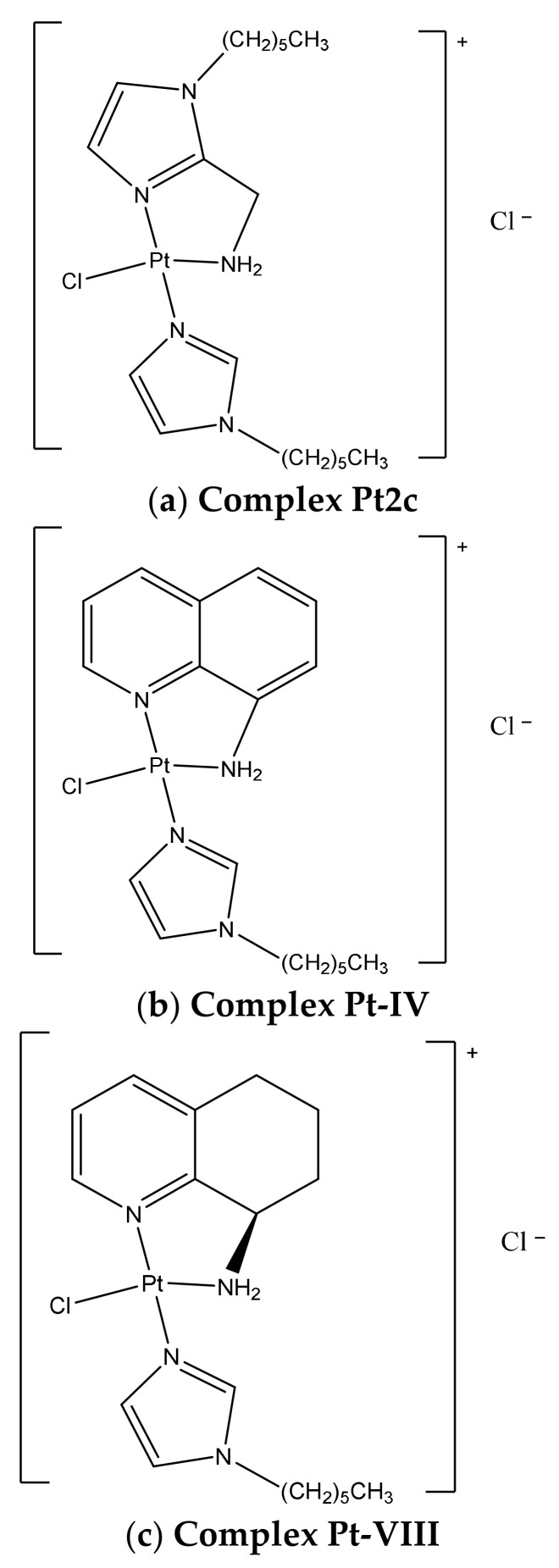
Chemical structures of the Pt compounds considered in this study: (**a**) complex Pt2c, (**b**) Pt-IV and (**c**) Pt-VIII.

**Figure 2 ijms-22-10551-f002:**
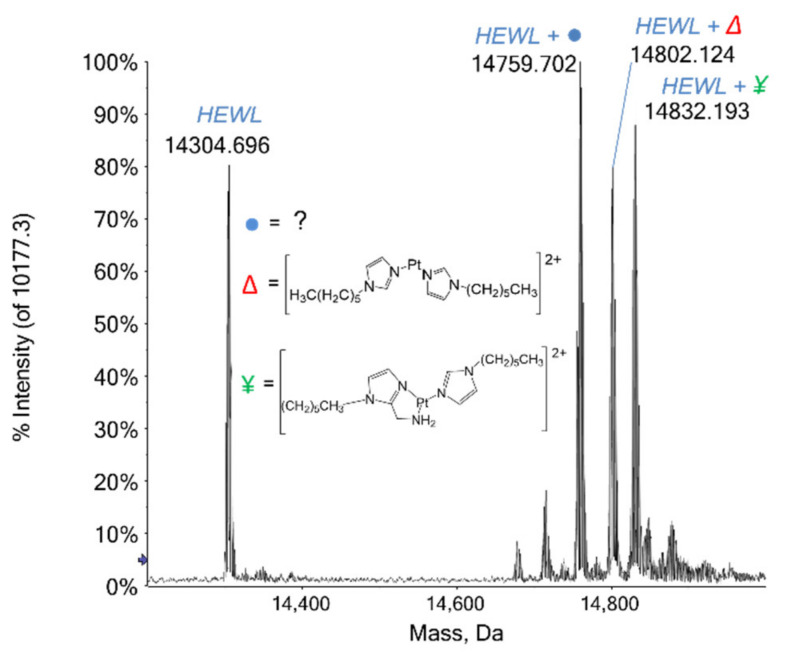
Deconvoluted mass spectra of HEWL (10^−7^ M) in 2 mM ammonium acetate solution at pH 6.8 incubated at 37 °C for 48 h with complex Pt2c in 1:3 protein-to-platinum compound molar ratio.

**Figure 3 ijms-22-10551-f003:**
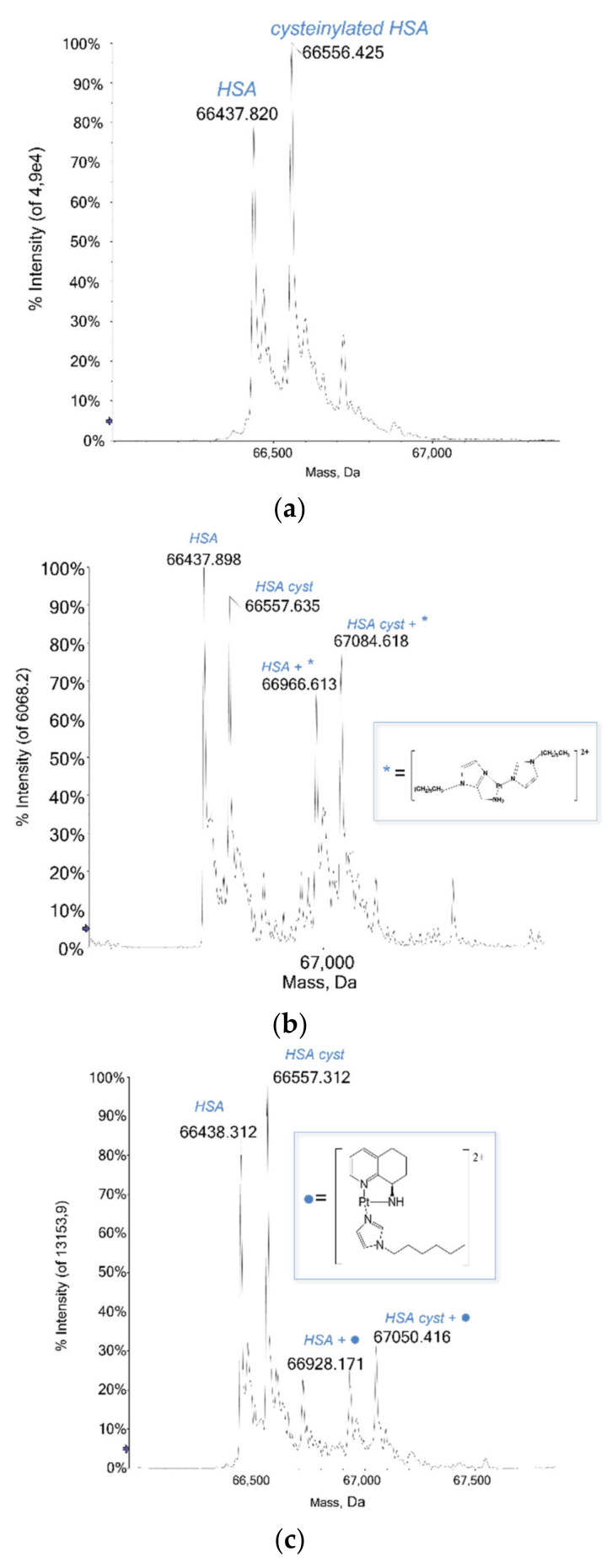
(**a**) Deconvoluted mass spectra of HSA (5 × 10^−7^ M) in 2 mM ammonium acetate solution at pH 6.8 incubated at 37 °C for 48 h with (**b**) Pt2c, [Pt((1-hexyl-1H-imidazol-2yl)(methanamine)(1-hexyl-1H-imidazole)]^2+^, (**c**) Pt-VIII, [Pt((R)-8-amino-5,6,7,8-tetrahydroquinoline)(1-hexyl-1H-imidazole)]^2+^ and (**d**) Pt-IV, [Pt((R)-8-aminoquinoline)(1-hexyl-1H-imidazole)]^2+^ in 1:3 protein-to-platinum compound molar ratio.

**Figure 4 ijms-22-10551-f004:**
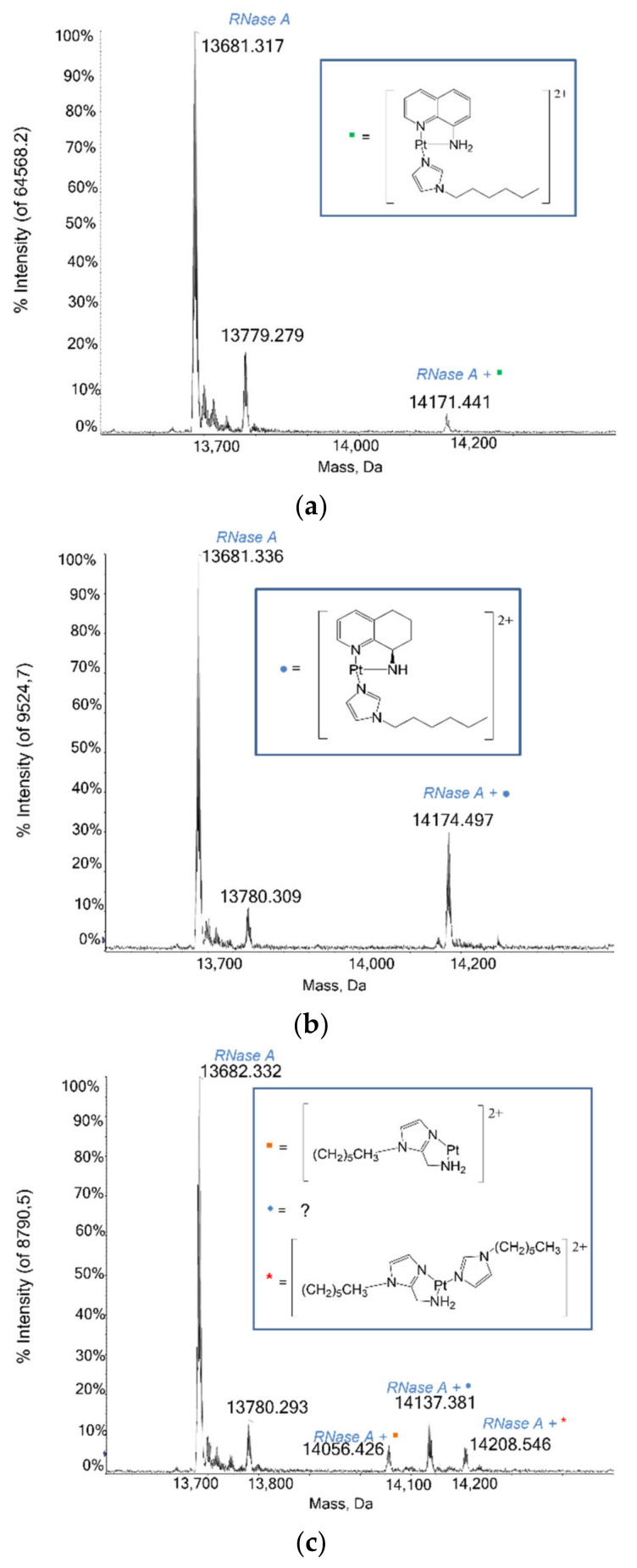
Deconvoluted mass spectra of RNase A (10^−7^ M) in 2mM ammonium acetate solution at pH 6.8 incubated at 37 °C for 24 h in 1:3 protein-to-platinum compound molar ratio with (**a**) Pt-IV, [Pt((R)-8-aminoquinoline)(1-hexyl-1H-imidazole)]^2+^, (**b**) Pt-VIII, [Pt((R)-8-amino-5,6,7,8-tetrahydroquinoline)(1-hexyl-1H-imidazole)]^2+^, (**c**) complex Pt2c, [Pt((1-hexyl-1H-imidazol-2yl)(methanamine)(1-hexyl-1H-imidazole)]^2+^.

**Figure 5 ijms-22-10551-f005:**
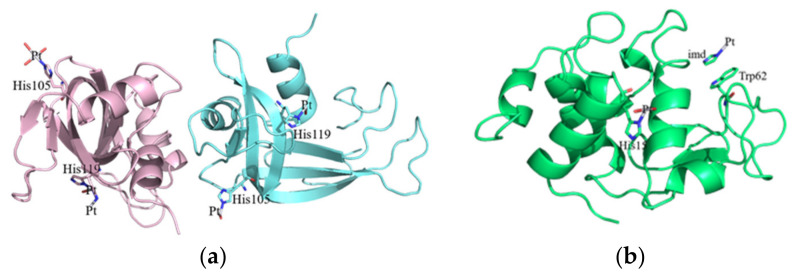
Overall structure of the adducts formed upon reaction of complex Pt2c with RNase A (**a**) and HEWL (**b**).

**Figure 6 ijms-22-10551-f006:**
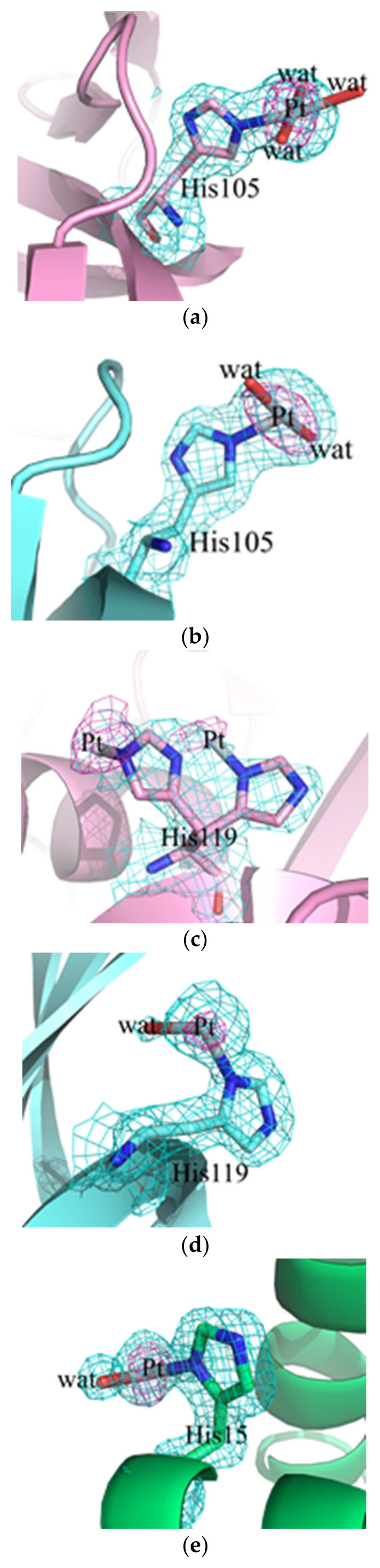
Pt binding sites close to the side chain of His105 (**a**,**b**) or His119 (**c**,**d**) in the adduct of RNase A with complex Pt2c and close to the side chain of His15 (**e**) and of Trp62 (**f**) in the adduct of the Pt complex with HEWL. 2Fo-Fc electron density maps are reported in cyan at 1.5 σ level. Anomalous difference map is reported in violet at 4.0 σ level.

**Figure 7 ijms-22-10551-f007:**
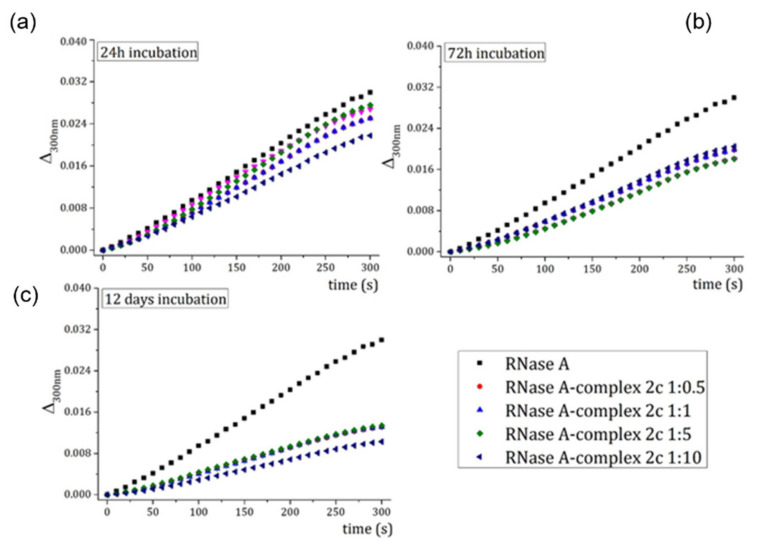
Hydrolysis of yeast RNA (evaluated by measuring the variation of absorbance at 300 nm as function of time upon addition of the protein to the yeast RNA sample) by native RNase A (black squares) and RNase A in the presence of different concentrations of complex 2c (colored triangles and circles). RNase A was incubated for 24 h (**a**), 72 h (**b**) and 12 days (**c**).

**Table 1 ijms-22-10551-t001:** Crystallographic parameters for the crystals of HEWL and RNase A adducts complex Pt2c.

	HEWL	RNase A
*Data collection*		
Soaking time	12 days	12 days
Space group	P43 21 2	C 2
a (Å)	77.20	100.14
b (Å)	77.20	31.79
c (Å)	38.53	73.74
α β γ (°)	90.0-90.0-90.0	90.0-90.3-90.0
Resolution range (Å)	34.52–1.22 (1.24–1.22)	28.04–2.13(2.17–2.13)
Observations	819,608 (42,148)	80,993 (3210)
Unique reflections	35,283 (1737)	13,153 (649)
Completeness (%)	100.0 (100.0)	99.0 (97.3)
Redundancy	23.2 (24.3)	6.2 (4.9)
†Rmerge (%)	0.056 (1.531)	0.078 (0.643)
Rmeas	0.057 (1.564)	0.086 (0.721)
Rpim	0.012 (0.315)	0.034 (0.318)
Average I/σ(I)	24.5 (2.2)	14.5 (2.3)
CC1/2	1.000 (0.861)	0.998 (0.898)
Anom. completeness (%)	100.0 (100.0)	98.1 (94.7)
Anom. Multiplicity	12.3 (12.6)	3.3 (2.6)
*Refinement*		
Resolution range (Å)	34.55–1.22	28.04–2.13
N. of reflections (working set)	32,310	12,454
N. of reflections (test set)		
R-factor/R-free (%)	12.87/16.97	19.75/27.43
N. of atoms	1276	2074
Average B-factors (Å2)		
All atoms	20.05	39.56
Pt atoms (occupancy)	0.20–0.25	0.20/0.25–0.35–0.50–0.55
R.m.s. deviations		
Bond lengths (Å)	0.019	0.013
Bond angles (°)	1.674	1.675
Ramachandran statistics (Validation Report)		
In preferred regions	113 (98.26%)	211 (93.36%)
In allowed regions	2 (1.74%)	10 (4.42%)
OutliersPDB code	0 (0.00%)7PNH	5 (2.21%)7PNI

†Rmerge= Σ hΣi |I(h,i)-<I(h)>|/ Σ hΣi I(h,i), where I(h,i) is the intensity of the ith measurement of reflection h and <I(h)> is the mean value of the intensity of reflection h.; Criteria used in determination of resolution cut.; Rpim <= 0.6000.; I/σI >= 2.00.; CC(1/2) >= 0.3000.

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
