# Peer review of "Reactions with Proteins of Three Novel Anticancer Platinum(II) Complexes Bearing N-Heterocyclic Ligands"

_ijms, 2021, doi:10.3390/ijms221910551_

Round 1

Reviewer 1 Report

Recommendation: Minor revision

Overview

With the general goal to propose non-conventional mode of action of anticancer Pt complexes potentially overcoming the classical side-effects of this kind of metallodrugs, the authors explore by ESI-MS, X-Ray and activity studies the reaction of three promising anticancer Pt (II) complexes, previously described, with proteins.

They show that the analyzed Pt complexes are able to form metallic adducts with three of the four model proteins. The type of adduct is different depending on both the Pt complex and the interacting protein. In two cases, crystallographic studies allow to precisely localize the Pt binding sites but the correlation between the type of complexes observed by MS and X-Ray is not guaranteed.

Overall, the experimental part is coherent and carefully performed, the results are clearly described and the paper well written.

The study is novel and add new elements in the context of the variegate patterns of reactivity of Pt complexes. 

Therefore, I recommend that a minor revision is warranted. I explain my concerns in more detail below. I ask that the authors specifically address each of my comments in their response.

Minor comments

  • in general, the quality of figures has to be improved, especially that of Figure 7 where the legend and axes are not clear.
  • lines 91-92: add references for this concept.
  • Figure 1: report names of molecules directly on the figure, it will help for direct identification of them during reading.
  • line 211-212: "degrades" should be better clarified.
  • line 304: explain why the ratio of Pf complex/protein is different in the case of HSA.
  • MS studies: the pattern of Pt complexes interactions with protein is variegate, however can you explain or comment on the total absence of reactivity with CytC or the minor reactivity of RNase compared to the other two proteins ? 
  • X-Ray studies: are the observed adducts covalent ? can you speculate about that point and also concerning the difference in between MS and crystallographic adducts ? 

Author Response

  • in general, the quality of figures has to be improved, especially that of Figure 7 where the legend and axes are not clear.

We have redrawn the figures, which are improved in the revised version of the manuscript

  • lines 91-92: add references for this concept.

References have been added

  • Figure 1: report names of molecules directly on the figure, it will help for direct identification of them during reading.

We added them as required

  • line 211-212: "degrades" should be better clarified.

We have replaced “degrades” with “breaks down”

  • line 304: explain why the ratio of Pt complex/protein is different in the case of HSA.

The sentence has been changed to “Then, aliquots of protein stock solution were mixed with aliquots of the platinum compounds at a protein-to-metal compound molar ratio of 1:3”

  • MS studies: the pattern of Pt complexes interactions with protein is variegate, however can you explain or comment on the total absence of reactivity with CytC or the minor reactivity of RNase compared to the other two proteins? 

The product of the reaction of a Pt compound with various proteins can be different for many reasons, since it depends on the nature of the protein (aminoacid sequence, protein dimension -in terms of number of protein residues and of protein folding-, accessibility of the potential binding sites, surface charge, pI and so on…)

  • X-Ray studies: are the observed adducts covalent ?

The adduct formed upon reaction of the Pt compound Pt2c with RNase A contains Pt centers directly bound to the side chains of His105 and His119 (coordinative bond). In the adduct with HEWL, both covalent and non-covalent interactions are observed. In the first binding site, the Pt center is coordinated by the side chain of His15, while in the second binding site the Pt compound stacks on the side chain of Trp62. This is clearly shown in Figure 6.

can you speculate about that point and also concerning the difference in between MS and crystallographic adducts? 

As discussed in the Discussion section “. It is possible that the fragments that are seen in the crystallographic studies can be different from those suggested by the ESI MS studies, due to the different experimental conditions and to the far longer incubation times of the crystallographic studies”

Reviewer 2 Report

The author's purpose of the investigation is very interesting, analyzing the reaction of 3 Pt compounds with model proteins. The paper is well written, globally clear and easy to read and understand. The methodology seems to be well dominated by the authors and the results are also very well defined and presented.  I would recommend the suggestions described below:

1)      At the end of the intro, it is not totally clear, at least in part, what is the main message and relevant points of the paper that should be emphasize at this stage. What are really the specific goals and findings of the paper? In the introduction section some references at the end of some periods are missing.

2) Discussion should be more assertive and concise and should follow the order of presentation of the paper. Eventually, it could be divided in sections highlighting the major topics analyzed in the paper.

3) The conclusion should resume partial and then global conclusions and also with perspectives for future research.

Author Response

  • At the end of the intro, it is not totally clear, at least in part, what is the main message and relevant points of the paper that should be emphasize at this stage. What are really the specific goals and findings of the paper? In the introduction section some references at the end of some periods are missing.

We corrected last paragraph and introduced a reference.

  • Discussion should be more assertive and concise and should follow the order of presentation of the paper. Eventually, it could be divided in sections highlighting the major topics analyzed in the paper.

We divided the discussion into two sections: the MS studies and the crystallographic ones.

  • The conclusion should resume partial and then global conclusions and also with perspectives for future research.

We introduced our goals and future perspectives in the last paragraph of the conclusion.